# Genetic Curriculum Learning for Distribution Generalization on the Travelling Salesman Problem

**Michael Li**
University of Washington
ml10872@uw.edu

**Christopher Haberland**
University of Washington
haberc@uw.edu

**Natasha Jaques**
University of Washington
nj@cs.washington.edu

## Abstract

The Travelling Salesman Problem (TSP) is a classic NP-hard combinatorial optimization task with numerous practical applications. Classic heuristic solvers – and Large Language Models (LLMs) using such solvers as a tool – can attain near-optimal performance for small problem instances, but become computationally intractable for larger problems. Real-world logistics problems such as dynamically re-routing last-mile deliveries demand a solver with fast inference time, which has led to specialized neural network solvers being favored in practice. However, neural networks struggle to generalize beyond the synthetic data they were trained on. In particular, we show that there exist TSP distributions that are realistic in practice, which also consistently lead to poor worst-case performance for existing neural approaches. To address distributional robustness, we present Genetic Curriculum Learning (GCL), an efficient novel approach utilizing automatic curricula. We also present TSPLib50, a dataset of realistically distributed TSP samples, which tests real-world distribution generalization ability without conflating this issue with TSP instance size. We evaluate our method on various synthetic datasets as well as TSPLib50, and compare to state-of-the-art LLM results and neural baselines. We demonstrate that GCL improves distributional robustness, with most of its performance gains coming from worst-case scenarios.

## 1 Introduction

From least-cost shipping and warehouse logistics to efficient automated circuit board drilling, the Traveling Salesman Problem (TSP) has an outsized impact on global trade, accounting for billions of dollars worth of saved time, energy, and harmful emissions. The TSP is NP-hard, which means there exists no efficient algorithm for finding exact solutions. Classic heuristic methods have prohibitive runtimes for real-world situations requiring fast and dynamic decision-making. Neural combinatorial optimization (NCO) methods seek to effectively solve the TSP at lower computational cost [18, 10, 22], but generalize poorly to unfamiliar distributions [11, 6]. In practice, such planning faults can be very expensive in terms of wasted time, money, and human resources.

Given impressive recent gains in reasoning capabilities of Large Language Models (LLMs) [27, 28, 17], LLMs potentially provide a promising path for solving novel TSP instances. However, LLMs currently perform suboptimally on the TSP [13, 26]. In this paper we directly study the performance of state-of-the-art LLMs prompted to solve TSP problems, and find that they have similarly prohibitive inference times as classic heuristics, in addition to inconsistent performance.

Instead, we propose a novel adversarial training technique to enhance the robustness of NCO methods. Rather than training on limited TSP datasets or randomly generated TSP instances, which is inefficient and wasteful due to the high-dimensional parameter space, we propose to use a curriculum learning approach in which environments and tasks are adaptively evolved to be more challenging [3]. As applied to TSP solvers, a "task" or a "level" would be a TSP instance that needs to be solved.

38th Conference on Neural Information Processing Systems (NeurIPS 2024).

Curriculum-based methods have shown promise for combinatorial optimization [15], but past works utilize very simple heuristic regimes, and are also focused on generalization across lengths, not distributions [20]. Zhang et al. [29] proposed a hardness-adaptive curriculum (HAC), which uses gradient ascent to produce increasingly harder levels. However, we show that HAC's sampling and gradient ascent procedure causes unreliable performance on specific types of TSP instances which are of practical interest.

In this paper, we seek to improve model robustness to different distributions. We present an NCO optimization approach which maintains the computational benefits of NCO methods while improving generalization on disparate but practical distributions. We make the following contributions: 1) We propose the TSPLib50 dataset, a testing dataset of 10,000 instances sampled from realistic distributions, designed to test the robustness of TSP solvers; 2) We propose an automatic curriculum which mutates high-improvement-potential training distributions; 3) We present empirical results comparing the performance to the best prior work on curriculum learning for NCO and state-of-the-art LLMs, and show that our method gives better worst-case performance and improved robustness to varying distributions of practical interest. Our method is also relatively efficient to train, requiring only a single GPU and no more than a few hours of training for each model.

## 2   Background

**Traveling Salesman Problem.** The Traveling Salesman Problem (TSP) is a NP-Complete combinatorial optimization problem (COP), which requires finding the shortest tour through a set of cities. The TSP has been of intense interest to computational theorists due to its applicability in many practical scenarios, especially in the logistics sector. Past works as well as this paper consider the 2D-Euclidean TSP, which is formally defined in the Appendix.

**Deep and Reinforcement Learning for TSP.** Neural combinatorial optimization (NCO), or the use of deep learning for combinatorial optimization, can be broadly grouped into three primary approaches: solutions utilizing 1) pointer networks [23, 14], 2) graph neural networks [8, 19, 30], or 3) transformers [12]. Reinforcement learning (RL) has seen successful applications in learning to solve the TSP [16, 4]. Deep RL methods often use a neural network to generate a tour, and then treat tour length as a negative reward, or "cost". Kool et al. [10] propose a transformer-based solver trained with REINFORCE [25], using a simple deterministic greedy rollout baseline. However, neural networks are known to often generalize poorly to distributions outside their training data, and existing NCO solvers are no exception. This makes them a risky solution for real-world deployments, in spite of their fast inference time. In this paper, we aim to improve the reliability and robustness of RL-based NCO approaches.

**Curriculum Learning for TSP.** Curriculum methods improve robustness and sample efficiency by proposing tasks to learn from which are optimal for learning by being neither too easy nor too hard [3, 24, 1], and have been applied to real-world problems such as web navigation [7]. In the context of Neural COP solvers, Zhang et al. [29] propose a hardness-adaptive curriculum (HAC) for the TSP, which mainly consists of two components: a hardness-adaptive generator that conducts gradient ascent on training instances, and a re-weighting procedure for batch gradients in favor of updates for harder levels. We directly compare to HAC in this work, and include the HAC hardness metric $\mathcal{H}(\mathbf{X}, M)$ as defined by Zhang et al. [29] in the Appendix. The hardness-adaptive generator conducts gradient ascent on input samples $\mathbf{X}^{(t)}$ given a model $M$ [29]: $\mathbf{X}^{(t)'} = \mathbf{X}^{(t)} + \eta \nabla_{X^{(t)}} \mathcal{H}(X^{(t)}, M)$.

## 3   Preliminary Study

**TSPLib50 and Other Evaluation Datasets.** We first motivate the creation of TSPLib50, a new testing dataset. TSPLib, a collection of real-world TSP instances, is often used as a benchmark for combinatorial optimization solvers [21]. Because TSPLib is based on real data, its distributions are both varied and relevant for real-world applications. However, many solvers are trained on relatively small TSP instances. When tested on TSPLib, the gaps incurred by such models are correlated with instance size. For instance, we find a strong Pearson correlation of 0.907 between TSPLib instance size and optimality gap of HAC models (see the Appendix). Improving model generalization to larger instance sizes often requires extensive computational resources, and is beyond the scope of preceding papers as well as this paper.

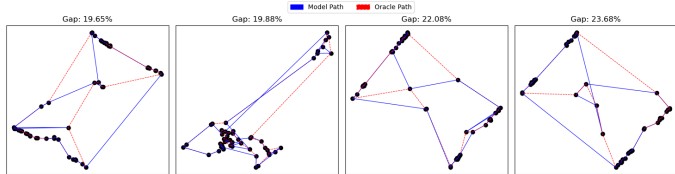

Figure 1: Example high-gap instances of a HAC model tested on TSPLib50. We see that all of these failure cases have large distances between node clusters, and thus deviate far from uniform levels.

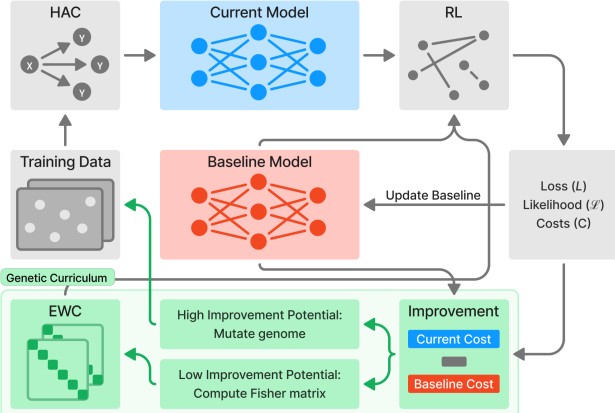

Figure 2: Architecture of our proposed Genetic Curriculum system. After the forward pass, we compute improvement and then mutate high-improvement levels while saving Fisher information about low-improvement levels.

Following the work of Zhang et al. [29], we focus on 50-node instances. Hence, we introduce TSPLib50, a dataset of 10,000 instances, each created by sampling 50 points uniformly at random from a TSPLib instance. Because the distribution of points in TSPLib50 is the same in expectation as the distribution of points in TSPLib, we can thus disentangle generalization ability on different *distributions* with generalization ability on different instance sizes.

We also test performance of our method on challenging synthetic distributions. We test on a Gaussian mixture distribution from prior work [29], and a "Diagonal" distribution of our design where all points align with a main diagonal. We justify and visualize these distributions in the Appendix.

**Hardness-Adaptive Curriculum Shortcomings.** While HAC improves performance by training on harder distributions [29], it only conducts one step of gradient ascent on data sampled from a uniform distribution. As a result, HAC fails to cover instances that deviate far from a uniform distribution.

In HAC, changes in $\mathbf{X}^{(t)}$ are determined by $\eta \nabla_{X^{(t)}} \mathcal{H}(X^{(t)}, M)$. We find that elements in $\eta |\nabla_{X^{(t)}} \mathcal{H}(X^{(t)}, M)|$ tend to have a mean around 0.077 and median around 0.023, which are small relative to the unit square $[0, 1]^2$ that points are placed in. Thus, points are only mildly perturbed.

In Figure 1, we visualize high-gap TSPLib50 levels for HAC, and find that HAC performs sub-optimally on levels with large distances between nodes. TSPLib50 bootstraps from real-world distributions, and thus represents use cases of practical interest. We seek to address this issue.

## 4 Genetic Curriculum Learning

**Improvement Potential Metric.** In Genetic Curriculum learning (GCL), we compute the "improvement potential" $I(\mathbf{X}, M)$ for each training instance after each epoch with the current model $M$ and REINFORCE baseline model $M'$: $I(\mathbf{X}, M) = \mathcal{C}_M(\mathbf{X}) - \mathcal{C}_{M'}(\mathbf{X})$. Note that $I(\mathbf{X}, M)$ is similar to $\mathcal{H}(\mathbf{X}, M)$ as used by Zhang et al. [29].

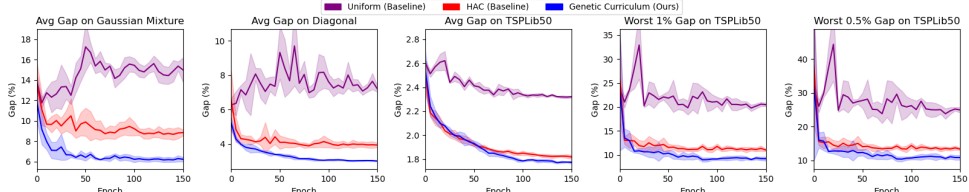

Figure 3: Gaps across training epochs of our proposed Genetic Curriculum model compared to baselines, on average cases across different distributions and worst-case scenarios in TSPLib50. The optimality gap of GPT-4o on TSPLib50 is around 90% on a small sample, and is not plotted due to the different scale of those values relative to existing results.

**Genetic Curriculum Algorithm.** GCL proposes novel usage of an evolutionary approach to maintain a population of challenging levels, drawing inspiration from genetic programming [2]. GCL stores a population of level "genes" that describe the probabilistic process creating the levels. After each epoch, the 50% of highest improvement-potential levels have their genes edited and placed into the next epoch. We find that mutating a population of genes achieves better results than mutating a population of levels. The genome consists of 6 bases: 0) Cluster size of the distribution points are drawn from; 1) Cluster width of the distribution points are drawn from; 2) Rotation angle; 3) Scale factor; 4) $x$-axis translation factor; 5) $y$-axis translation factor. Through this genome, we try to address distribution invariance, rotational invariance, scale invariance, and translation invariance. Technical details of level sampling from genomes, genetic mutation procedure, and motivation for related hyperparameters are in the Appendix.

GCL also uses Elastic Weight Consolidation (EWC) [9] to maintain performance on its learned knowledge, because as the genetic curriculum evolves to harder instances, it is possible that catastrophic forgetting is leading the model to perform poorly on easier instances. Figure 2 provides a diagram of the architecture of GCL. An algorithmic specification of GCL is provided in the Appendix.

## 5 Experiments

Our experiments work on fine-tuning an attention-based model with a REINFORCE rollout baseline, previously trained exclusively on uniform random distributions. We compare our model to results from OpenAI's GPT-4o, a state-of-the-art LLM. We also compare against 2 NCO baselines: a "Uniform" baseline which samples from uniform distributions without curriculum, and a "HAC" baseline which samples from uniform distributions but uses HAC. Notably, all experiments are run on a single GPU, and no model takes longer than a few hours to train. Full experimental details and hyperparameters are in the Appendix.[1]

We plot gaps of all models relative to oracles. We present results for average gaps on the Gaussian Mixture, Diagonal, and TSPLib50 distributions. We also present results on worst-case 1%, 0.5%, and 0.1% of gaps on TSPLib50, to demonstrate robustness to challenging out-of-distribution cases.

To further investigate our method, we run three tests to better interpret GCL. First, we run ablation tests on the genome and EWC components of our method. Second, we plot the distribution of each genome base over the course of training, to better understand the role the genome plays in GCL. Third, we plot the optimality gap of our baseline model $M'$ and current model $M$ on training data over the course of training, to better understand model convergence behavior.

## 6 Results

**Large Language Model Performance.** Despite advances in mathematical and reasoning capabilities, Large Language Models (LLMs) often fail to find satisfactorily optimal TSP solutions, and have prohibitively slow inference, requiring around 47 seconds on average. Even with prompt engineering,

---

[1]All code is provided at `https://github.com/ML72/Genetic-Curriculum-TSP/`

LLMs still produce inconsistent and suboptimal responses. Details about our TSP-related LLM experiments are in the Appendix.

| Dataset | Model | Gap Avg (%) | Gap Std (%) |
|---------|-------|-------------|-------------|
| Gaussian Mixture | Uniform | 15.0049 | 1.1970 |
| | HAC | 8.8460 | 0.3032 |
| | **GCL (Ours)** | **6.2214** | **0.2155** |
| Diagonal | Uniform | 7.2115 | 0.1822 |
| | HAC | 3.9447 | 0.1346 |
| | **GCL (Ours)** | **3.0165** | **0.0318** |
| TSPLib50 | Uniform | 2.3206 | 0.0082 |
| | HAC | 1.8183 | 0.0167 |
| | **GCL (Ours)** | **1.7738** | **0.0119** |

Table 1: Average Model Gap Across Distributions

**Average Gaps.** Average gap results can be seen in Table 1. On all distributions, HAC already improves significantly on the uniform baseline, as HAC uses a hardness-adaptive generator. GCL, our proposed method, achieves consistent improvement over HAC on the harder distributions. We observe an approximately 1/4 factor gap decrease on both hard distributions: the gap decreases from 8.85% to 6.22% on Gaussian mixtures, and from 3.94% to 3.02% on the diagonal distribution. For the TSPLib50 distribution, GCL improves only slightly on HAC in terms of average gap. This makes sense because a large portion of TSPLib50 levels are easy, while GCL focuses on robustness to challenging levels.

We find decreases in average gap between HAC and GCL to be statistically significant, with values of $p < 0.01$ in two-sample t-tests for all distributions.

**Worst-Case Gaps.** We can see that the slight improvement on TSPLib50 average gap mostly comes from gap decreases on hard levels by analyzing worst-case scenarios. On TSPLib50, GCL provides a 1.82% gap improvement on the worst 1% of cases, a 2.52% gap improvement on the worst 0.5%, and a 3.42% gap improvement on the worst 0.1%. This is significant because in large-scale real-world applications that route millions of TSP problems every day, 1% of routes is still an important and costly fraction. For example, if a large shipping company routes 40,000 loads, 1% of routes would still equate to 400 loads.

We find decreases in worst-case TSPLib50 gap between HAC and GCL to be statistically significant, with values of $p < 0.001$ in two-sample t-tests for worst 1% and worst 0.5%, and $p < 0.03$ for worst 0.1%. GCL also improves gap from 204.40% by HAC to 98.47% on the worst 1% of Gaussian Mixture cases, and from 31.63% by HAC to 14.03% on the worst 1% of Diagonal cases. This demonstrates GCL's significant impact on improving robustness in the most challenging scenarios. Detailed tables are in the Appendix.

**Method Interpretations.** In our ablation tests, we find that performance gains are mainly provided by the genetic component of our method. In our genome evolution plots, we observe that the diversity of genome bases decreases over training, suggesting our genome is effective at encouraging exploration of new configurations. In our optimality gap plot, we observe that the baseline gap starts higher than model gap, but dramatically decreases at around epoch 60, which further supports the previous point on the genome promoting exploration. Detailed results and explanations are in the Appendix.

## 7 Discussion

Our results demonstrate that GCL is able to significantly improve the performance of TSP solvers on hard distributions. We also show that a portion of this improvement occurs in "worst-case" scenarios on real-world distributions of practical interest. Such improved robustness and performance guarantees are significant in real-world deployment.

GCL is a general methodology, and could be applied to other NCOs methods or COPs. As the Kool et al. [10] architecture generalizes to other problems such as the Vehicle Routing Problem (VRP) and Capacitated VRP (CVRP), it would be exciting to see GCL used for other COPs.

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

## A  Formal Definitions

### A.1  Traveling Salesman Problem

Formally, given a set of cities $V = \{1, 2, \ldots, n\}$ and a $n \times n$ distance matrix $D$ where $D_{i,j}$ is a real number denoting the distance between city $i$ and city $j$, the Traveling Salesman Problem (TSP) seeks to find the optimal permutation of cities $\sigma^*$ that minimizes total tour length [5]:

$$\sigma^* = \arg\min_{\sigma} \left[ D_{\sigma(n),\sigma(1)} + \sum_{i=1}^{n-1} D_{\sigma(i),\sigma(i+1)} \right] \tag{1}$$

The 2D-Euclidean TSP is a special case of the TSP where all cities are given a position on the 2D Euclidean plane, and all $D_{i,j}$ represent the Euclidean distance between cities $i$ and $j$. Because $\sigma^*$ is theoretically translation-invariant and scale-invariant, 2D-Euclidean TSP problems often provide city locations that are translated and scaled to fit in the $[0, 1]^2$ unit square.

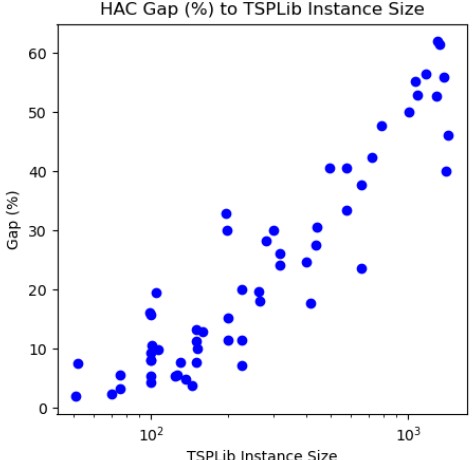

Figure 4: There is a Pearson correlation of $0.907$ between HAC gap and TSPLib instance size. All 2D-Euclidean TSPLib instances with 1400 or fewer nodes are included.

## A.2 Elastic Weight Consolidation

For an old task $A$ and a new task $B$, Elastic Weight Consolidation (EWC) computes a mean given by the model parameters $\theta_A^*$ and the diagonal of the Fisher information matrix $F$. For some importance hyperparameter $\lambda$ and loss $\mathcal{L}_B(\theta)$ on task $B$, EWC loss is formally defined as follows [9]:

$$\mathcal{L}(\theta) = \mathcal{L}_B(\theta) + \sum_i \frac{\lambda}{2} F_i (\theta_i - \theta_{A,i}^*)^2 \tag{2}$$

## A.3 Optimality Gap

The optimality gap $\mathcal{G}$ for a dataset $\mathbf{X}$ used by the hardness-adaptive generator is the gap in cost $\mathcal{C}$ between the current model $M$ and an oracle model $M^*$:

$$\mathcal{G}(\mathbf{X}, M) = \frac{\mathcal{C}_M(\mathbf{X}) - \mathcal{C}_{M*}(\mathbf{X})}{\mathcal{C}_{M*}(\mathbf{X})} \tag{3}$$

## A.4 HAC Hardness Metric

The hardness metric $\mathcal{H}$ for a dataset $\mathbf{X}$ used by the hardness-adaptive generator is the gap in cost $\mathcal{C}$ between the current model $M$ and a surrogate model $M'$ which is greedily updated by a few steps of gradient descent [29]:

$$\mathcal{H}(\mathbf{X}, M) = \frac{\mathcal{C}_M(\mathbf{X}) - \mathcal{C}_{M'}(\mathbf{X})}{\mathcal{C}_{M'}(\mathbf{X})} \tag{4}$$

Note that the formulation for $\mathcal{H}(\mathbf{X}, M)$ is similar to the formulation for $\mathcal{G}(\mathbf{X}, M)$. In fact, Zhang et al. [29] observe that $\mathcal{H}(\mathbf{X}, M)$ is always a lower bound for $\mathcal{G}(\mathbf{X}, M)$.

# B Preliminary Study

## B.1 Evaluation Datasets

Figure 4 demonstrates the high correlation between TSPLib instance size and resulting performance, justifying the necessity of our creation of TSPLib50.

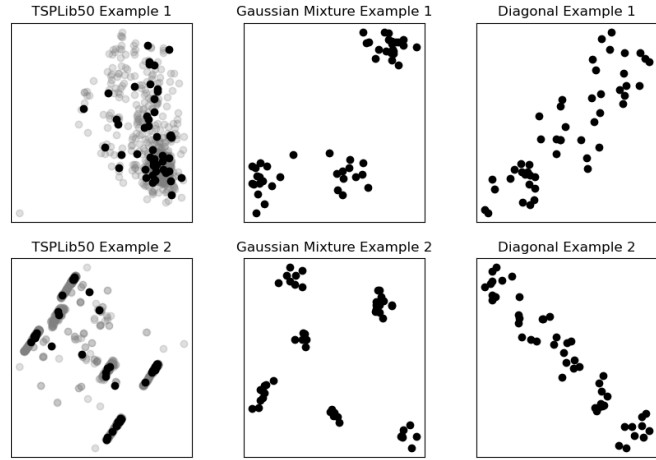

Figure 5: Visualizations of example instances from the distributions we evaluate on. For TSPLib50, we also plot the original TSPLib instance that we sampled from.

---

**Algorithm 1** Generate TSPLib50 Dataset

---
**Parameter**: Dataset size $S$
**Output**: Dataset $D$

 1: Initialize empty dataset $D$
 2: Initialize list of 2D-Euclidean TSPLib levels $L$
 3: $L \leftarrow [l \text{ in } L \text{ if size}(l) \geq 150]$
 4: **for** $i = 0, 1, \ldots, S - 1$ **do**
 5:     $l \leftarrow L[i \% \text{length}(L)]$
 6:     $l' \leftarrow 50 \text{ points} \sim l$
 7:     $D \leftarrow D + l'$
 8: **end for**
 9: **return** $D$

---

We also test on the Gaussian mixture distribution because it tends to pose a challenge to existing TSP solvers, as noted by previous works [29].

We also test on a "Diagonal" distribution of our design because it is intended to be difficult in another manner. Previously, we identified a common feature of HAC failure cases being that they have much empty space, and we justified this interpretation mathematically. However, another common feature of those cases is that they have points in distinct clusters. The diagonal distribution aims to experimentally demonstrate that empty space is a primary factor for difficulty, by having all points aligned along a main diagonal. As such, there is only one cluster, but there is still much empty space on the level.

Figure 5 visualizes the TSPLib50, Gaussian Mixture, and Diagonal distributions.

## B.2 Distribution Generation

Our algorithm for generating TSPLib50 is specified in Algorithm 1. Note that when generating TSPLib50, we only sample from TSPLib instances with 150 or more nodes to ensure that there is sufficient diversity in the generated TSPLib50 instances.

Our algorithm for generating Gaussian mixture datasets is specified in Algorithm 2. Note that while we generate Gaussian mixture distributions in the same fashion as Zhang et al. [29], our dataset composition is different. Zhang et al. [29] add instances of the uniform distribution and Gaussian mixtures with $c_{dist} = 1$ to the testing dataset. Note that Gaussian mixtures with $c_{dist} = 1$ are "close to uniform" by our definition, as there is not much empty space on the level. We do not do this, and keep the Gaussian mixture distribution as purely Gaussian mixtures. Thus, our reported gaps

---

**Algorithm 2** Generate Gaussian Mixture Dataset

---

**Parameter**: Dataset size $S$

**Output**: Dataset $D$

1:  Initialize empty dataset $D$
2:  **for** $c_{dist} \in [10, 20, 30, 40, 50, 60, 70, 80, 90, 100]$ **do**
3:      **for** $i = 0, 1, \ldots, S/10 - 1$ **do**
4:          Number of centers $\sim \text{Unif}(3, 6)$
5:          Number of points per center $\sim$ Multinomial
6:          Place centers within $[0, c_{dist}]^2$
7:          $l \leftarrow 50$ points $\sim \mathcal{N}(\text{centers}, 2I_2)$
8:          Rescale $l$ to fit in $[0, 1]^2$
9:          $D \leftarrow D + l$
10:     **end for**
11: **end for**
12: **return** $D$

---

---

**Algorithm 3** Generate Diagonal Dataset

---

**Parameter**: Dataset size $S$

**Output**: Dataset $D$

1:  Initialize empty dataset $D$
2:  **for** $d \in [1, 2, 3, 4, 5]$ **do**
3:      **for** $i = 0, 1, \ldots, S/5 - 1$ **do**
4:          $l \leftarrow 50/d$ points distributed uniformly at each location from $(1, 1), (2, 2), \ldots (d, d)$
5:          Negate $y$-coordinates in $l$ with $p = 0.5$
6:          Rescale $l$ to fit in $[0, 1]^2$
7:          $D \leftarrow D + l$
8:      **end for**
9:  **end for**
10: **return** $D$

---

with HAC are higher than those reported by Zhang et al. [29], as uniform distributions and Gaussian mixtures with $c_{dist} = 1$ incur very low gap.

Our algorithm for generating Diagonal datasets is specified in Algorithm 3.

## C  Genetic Curriculum Learning

### C.1  Algorithmic Specification and Architecture

Algorithm 4 provides an algorithmic specification of GCL.

### C.2  Genetic Algorithm

Recall that the GCL genome consists of 6 bases: 0) Cluster size of the distribution points are drawn from; 1) Cluster width of the distribution points are drawn from; 2) Rotation angle; 3) Scale factor; 4) $x$-axis translation factor; 5) $y$-axis translation factor.

Levels are drawn from a "clustered uniform distribution", and then rotated, scaled, and translated in that order. Combined, these parameters address distributional invariance, rotational invariance, scale invariance, and translation invariance. Intuitively, the clustered uniform distribution generates points in various clusters, which serves as a better starting point for the gradient ascent step in the hardness-adaptive generator to reach a variety of distributions. A sampling algorithm for this distribution is specified in Algorithm 5.

Beginning each epoch, training data is sampled from the genomes in a probabilistic process. Our algorithm for sampling a level $l$ from a gene $g$ is specified in Algorithm 6.

---
**Algorithm 4** Genetic Curriculum Learning
---
**Input**: Current model $M$, baseline model $M'$, hardness-adaptive generator $\phi$, genetic mutation procedure $\psi$, genome distribution $\Psi$
**Parameter**: Batch size $B$, training epochs $L$, EWC sample size $N$, EWC importance $\lambda$
**Output**: Fine-tuned model $M'$

  1: Initialize and warm up $M$ and $M'$
  2: Initialize genome $G \sim \Psi$
  3: **for** $i = 1, 2, \ldots, L$ **do**
  4:     Sample dataset $D \sim G$
  5:     $D' \leftarrow \phi(D)$
  6:     **for** $b = 1, 2, \ldots, |D|/B$ **do**
  7:         Get batch data $\{\mathbf{X}\}_{i=1}^{B}$ from $D$
  8:         Pass batch data through baseline model $M'$
  9:         Pass batch data through model $M$
 10:         Update model parameters with weighted gradients
 11:     **end for**
 12:     Compute improvement $I = \mathcal{C}_M(\mathbf{X}) - \mathcal{C}_{M'}(\mathbf{X})$
 13:     Sort $D$ and $G$ using $I$
 14:     Compute EWC Fisher matrix $F$ with $D[0 : N]$
 15:     $G[|G|/2 : |G|] \leftarrow \psi(G[|G|/2 : |G|])$
 16:     $G[0 : |G|/2] \sim \Psi$
 17:     **if** $\mathcal{C}(M) < \mathcal{C}(M')$ **then**
 18:         $M' \leftarrow M$
 19:     **end if**
 20: **end for**
 21: **return** $M$
---

---
**Algorithm 5** Generate Clustered Uniform Dataset
---
**Parameter**: Dataset size $S$, cluster size $c$, noise $\epsilon$
**Output**: Dataset $D$

  1: Initialize empty dataset $D$
  2: **for** $i = 0, 1, \ldots, S - 1$ **do**
  3:     Number of centers $\leftarrow 50/c$
  4:     Place centers within $[0, 1]^2$
  5:     $l \leftarrow 50$ points $\sim \text{Unif}(\text{centers} - \epsilon, \text{centers} + \epsilon)$
  6:     Rescale $l$ to fit in $[0, 1]^2$
  7:     $D \leftarrow D + l$
  8: **end for**
  9: **return** $D$
---

**Genetic mutation procedure $\psi$:** After each epoch, we select the 50% of highest-improvement genes and mutate their bases with $\psi$. Each base is incremented with probability 1/12 and decremented with probability 1/12. Note that in aggregate, this means each base is mutated with probability 1/6. Following are the min/max bounds and increment/decrement magnitudes for the bases:

- 0: Min = 1, Max = 25, Inc/Dec = $\pm 1$
- 1: Min = 0.03, Max = 0.08, Inc/Dec = $\pm 0.01$
- 2: Min $\approx -2\pi$, Max $\approx 2\pi$, Inc/Dec = $\pm 0.1$
- 3: Min = 0.7, Max = 1, Inc/Dec = $\pm 0.1$
- 4: Min = 0, Max = 1, Inc/Dec = $\pm 0.1$
- 5: Min = 0, Max = 1, Inc/Dec = $\pm 0.1$

**Genome distribution $\Psi$:** After each epoch, we select the 50% of lowest-improvement genes and resample them from $\Psi$, as specified below per base. These initial values are often "middle" values that allow exploration in both directions, which mitigates the possibility of getting stuck in a gene pool "local minima", as detailed in our hyperparameter interpretations below.

---
**Algorithm 6** Sample Level from Genes
---
**Input:** Gene $g$
**Output**: Level $l$
 1: $l \sim \text{ClusteredUniform}(c = g[0], \epsilon = g[1])$
 2: $l \leftarrow \text{Rotate}(l, \theta = g[2])$
 3: Rescale $l$ to fit in $[0, 1]^2$
 4: $l \leftarrow l \times g[3]$
 5: $l \leftarrow l + (\Delta x = g[4](1 - g[3]), \Delta y = g[5](1 - g[3]))$
 6: **return** $l$
---

- 0: Uniform split between $\{1, 5, 10, 15\}$
- 1: 0.05
- 2: 0
- 3: 1
- 4: 0.5
- 5: 0.5

Following are interpretations for the two important end-of-epoch mutation hyperparameters:

- **Mutate 50% of high-improvement levels:** We mutate the 50% of most improved levels and re-sample 50% as it strikes a balance between fresh training distributions and hard distributions that the model needs to improve at. This also minimizes the risk of getting stuck in possible gene pool "local minima", where the majority of genes are focused on distributions that used to be hard, but are no longer challenging.

- **Mutate each genome base with 1/6 probability:** During mutation, each base is mutated with 1/6 probability; because there are 6 bases in each level's genome, in expectation only one base is modified per mutation. This allows genetic diversity of levels while preventing levels from mutating so much that the mutated genes' difficulty is vastly different from the original genes' difficulty.

### C.3 Elastic Weight Consolidation

After each epoch, a number $N$ of least-improved levels are saved and used to compute Fisher information diagonals and means for the network parameters $\theta$. This is then used in the EWC penalty for constraining gradient updates in the next epoch. The intuition is that low improvement-potential levels are likely representative of the model's strengths, and thus important related parameters should not be changed.

## D  Experiments

### D.1  Setup

Following the work of Kool et al. [10], we use an attention-based architecture trained with a REIN-FORCE rollout baseline. We also use the gradient re-weighting curriculum from Zhang et al. [29]. All our experiments work on fine-tuning a model previously trained exclusively on uniform random distributions. We work off the Kool et al. [10] codebase, which is released under an MIT license.

We compare our model against two baseline methods: a uniform baseline trained on uniform distributions without any form of curriculum, and a HAC baseline which samples from uniform distributions but uses HAC to make training instances more challenging. Note that the uniform baseline is equivalent to the model used by Kool et al. [10], while the HAC baseline is equivalent to the model used by Zhang et al. [29].

We plot gaps of all models relative to Concorde solutions, as Concorde is an optimal solver. For all distributions, 10,000 instances are sampled for evaluation. We also train 5 models for each setting, and report averaged results between the 5 models for each setting. We assume a normal distribution of error across these averages, and plot error bars equal to 1-$\sigma$ of these averages.

### D.2 Hyperparameters

The base hyperparameters used for training all models are listed below. For fair comparison, we use the same model architecture used by Kool et al. [10] and Zhang et al. [29].

- **Architecture:** embedding dim = 128, hidden dim = 128, num encode layers = 3
- **Training:** graph size = 50, baseline = rollout, baseline warmup epochs = 0, epoch size = 65536, batch size = 1024, epochs = 151, LR decay = 0.98
- **HAC:** $\eta = 5$, adaptive percent = 100
- **EWC:** $\lambda = 1$, warmup epochs = 20, num samples = 2048

The exact training and evaluation commands which we use to obtain our results are included in our code. Those commands give additional information about default/implicit hyperparameters not listed.

Our criteria for selecting final parameter settings was best average gap on testing distributions. Note that we did not tune all parameters concurrently; in particular, our EWC parameters were selected at a stage of our tuning when EWC had a more significant effect on results. Hyperparameter ranges used for tuning are as follows:

- **Architecture:** Not tuned, consistent with Kool et al. [10] and Zhang et al. [29]
- **Training:** baseline warmup epochs $\in [0,1]$, epochs $\in [101, 251]$, LR decay $\in [0.95, 1]$
- **HAC:** Not tuned, consistent with Zhang et al. [29]
- **EWC:** $\lambda \in [0.01, 1000]$, warmup epochs $\in [0,50]$, num samples $\in [128, 2048]$

### D.3 Details

We train 5 models for each setting, and report aggregate results over the 5 models for each setting. All test datasets were generated with the random seed "1234". Our models are not explicitly seeded, but we find variance between runs to be consistent and reproducible.

All experiments were run on a singular NVIDIA L40 GPU on a Linux operating system with 20GB requested memory. However, our hyperparameter settings do not fully utilize GPU capabilities, and results should be reproducible on lower-memory GPUs such as the GeForce RTX 4060. No model takes longer than a few hours to train.

### D.4 Source Code

Our source code is publicly available at `https://github.com/ML72/Genetic-Curriculum-TSP`. Relevant software libraries and frameworks are listed in the dependencies file included in our code. We also include special documentation on installing pyconcorde, which we use as our oracle solver. All exact code used for plotting and conducting statistical tests is included in our code.

## E   Results

### E.1   Large Language Model Results

| Instance | Inference Time (s) | Optimal Distance | GPT-4o Distance | Gap (%) |
|----------|--------------------|------------------|-----------------|---------|
| Instance 1 | 52 | 4.8462 | 6.7730 | 39.7591 |
| Instance 2 | 50 | 5.5103 | 24.4655 | 343.9992 |
| Instance 3 | 41 | 4.0004 | 4.4332 | 10.8190 |
| Instance 4 | 47 | 3.8288 | 4.4590 | 16.4596 |
| Instance 5 | 47 | 3.0122 | 4.3116 | 43.1364 |
| **Average** | **47.4** | **4.2396** | **8.8885** | **90.8347** |

Table 2: GPT-4o Performance on Solving the TSP

We run brief experiments to demonstrate that large language models have inconsistent performance and lengthy inference times when asked to solve TSP problems. We prompted OpenAI's GPT-4o, which boasts state-of-the-art performance on mathematical and logical reasoning tasks, to solve the first 5 TSP instances in TSPLib50. Performance on these instances is printed in Table 2.

As opposed to prior work which uses meta-prompts [26], we prompt the LLM to directly solve a given TSP instance, for simplicity and faster inference. We use prompt engineering to improve response performance and consistency to the best of our ability. In our prompt, we ask for a permutation of the indices 1 to 50 and nothing else, to encourage the LLM to consistently return a valid permutation. We also ask the LLM to solve the problem to the best of its ability, to avoid errors where the LLM tries to use tools or libraries not in its environment and thus encounters an error. If we directly ask for a solution to the TSP instance without such further instructions, the majority of responses do not provide a tour, falling into one of the following failure cases:

- GPT-4o correctly identifies the problem as being the TSP and describes methods for approximating a solution, but makes no attempt to solve the instance provided.
- GPT-4o provides code which runs classic heuristic-based algorithms to solve the TSP, but does not provide an actual permutation.
- GPT-4o writes code and attempts to run it, but encounters into an error because some library that it needs is not in its current environment.

A sample prompt is printed below:

> There are 50 cities, respectively at the following locations on a 2D plane:
> (0.42731, 0.17487), (0.91628, 0.1503), (0.29484, 0.32477), (0.79805, 0.83944),
> (0.19764, 0.06484), (0.16905, 0.59943), (0.2539, 0.38879), (0.63631, 0.06185),
> (0.29277, 0.18652), (0.15599, 0.57022), (0.12723, 0.37378), (0.64433, 0.03523),
> (0.05164, 0.40614), (0.71752, 0.3749), (0.34851, 0.0), (0.53609, 0.61541),
> (0.41741, 0.57393), (0.47766, 0.1533), (0.25821, 0.15485), (0.31986, 0.75521),
> (0.44394, 0.63197), (0.84433, 0.39004), (0.33283, 0.56372), (0.325, 0.59029),
> (0.02606, 0.49288), (0.573, 0.78519), (0.4752, 0.01513), (0.40449, 0.54465),
> (0.23839, 0.11115), (0.23977, 0.06993), (0.4329, 0.01002), (0.92899, 0.40017),
> (0.21591, 0.14988), (0.07178, 0.56037), (0.88931, 0.31247), (0.25623, 0.01655),
> (0.24801, 0.0432), (0.16818, 0.30984), (0.45731, 0.88187), (0.80764, 0.22011),
> (0.44103, 0.49385), (0.62063, 0.62541), (0.72454, 0.86523), (0.36167, 0.76018),
> (0.55969, 0.02518), (0.42486, 0.14698), (0.01606, 0.74696), (0.20613, 0.54872),
> (0.06619, 0.72514), (0.80553, 0.81297)
> What is the optimal tour permutation of the cities to minimize the total distance traveled? Solve the problem to the best of your ability. Reply with only a permutation of the indices 1 to 50, and nothing else.

From Table 2, we can immediately see that the inference time required by GPT-4o is prohibitively expensive in situations requiring fast and dynamic decision-making. Furthermore, the performance is inconsistent: we can see that the gap incurred by GPT-4o has high variance, which is especially evident in the gap incurred on instance 2. Even in the best-case scenarios, the gap provided is still suboptimal compared to NCO methods. It is also worth mentioning that the formatting of the outputs produced by GPT-4o have slight inconsistencies as well. A permutation was provided for all 5 instances, but sometimes there were artifacts such as extra square brackets around the permutation. These small formatting variations pose a possible risk of feeding unexpected input into downstream applications that utilize these outputs.

### E.2 Worst-Case Gaps

Worst-case gap results on TSPLib50 can be seen in Table 3. We can observe that there is significant improvement by GCL on the hardest TSP instances.

### E.3 Ablations

In Table 4, we report ablation results for the two aspects of our curriculum. It can be seen that without the genome component, the gap increases significantly, while without the EWC component,

| % Worst | Model | Gap Avg (%) | Gap Std (%) |
|---|---|---|---|
| 1% | Uniform | 20.4826 | 0.3445 |
| | HAC | 11.0197 | 0.2942 |
| | **GCL (Ours)** | **9.2047** | **0.4223** |
| 0.5% | Uniform | 24.9234 | 0.7753 |
| | HAC | 13.3394 | 0.3584 |
| | **GCL (Ours)** | **10.8200** | **0.6798** |
| 0.1% | Uniform | 42.7885 | 4.6717 |
| | HAC | 19.4061 | 0.5314 |
| | **GCL (Ours)** | **15.9905** | **2.0829** |

Table 3: Worst Case Gap on TSPLib50

| Dataset | Model | Gap Avg (%) | Gap Std (%) |
|---|---|---|---|
| Gaussian Mixture | **GCL (Ours)** | **6.2214** | **0.2155** |
| | No EWC | 5.9941 | 0.1912 |
| | No Genome | 8.8520 | 0.7046 |
| Diagonal | **GCL (Ours)** | **3.0165** | **0.0318** |
| | No EWC | 3.0217 | 0.0344 |
| | No Genome | 3.8851 | 0.1026 |
| TSPLib50 | **GCL (Ours)** | **1.7738** | **0.0119** |
| | No EWC | 1.7792 | 0.0115 |
| | No Genome | 1.8232 | 0.0127 |

Table 4: Average Model Gap Ablations

performance is similar. We conclude that the genome component is most impactful for improving performance. We find a similar trend for ablation tests on TSPLib50 worst-case scenarios.

### E.4 Genome Evolution

To better understand how the genome evolves over training, we plot the distribution for each genome base over the course of training in Figure 6. We plot this for a single GCL run. Note that the high-frequency bright bands are the default values that new genome instances are sampled from. Recall that half of the genomes are resampled after each epoch, and only 1 base mutates in expectation per genome each epoch; this explains why the default values are considerably higher frequency than neighboring values.

However, outside of the default values, we can see that the genome diversity slowly decreases over training. This is most visible in the plots of bases 2, 4, and 5. We interpret this as indicating that the genome is effective: there is rapid evolution at first as the genome population explores new genome configurations, but this incentive for diversity decreases as the model eventually learns to solve these novel configurations.

### E.5 Gap Over Training

In Figure 7, we plot the optimality gap for our current model and our baseline model on training levels, over training. We plot this for a single GCL run. As it is too computationally intensive to compute the oracle for all levels in an epoch, we uniformly sample 1000 training levels from every 15th epoch, and calculate optimality gap on those levels.

We can see that notably, the baseline gap starts higher than the model gap, but then decreases significantly at around epoch 60. We also interpret this as a further indicator that the genome is effective: at first the baseline model struggles to keep up with the levels that the current model is training on due to rapid genome evolution, but over time the baseline learns to generalize to those new levels. This finding is consistent with our interpretation of Figure 6.

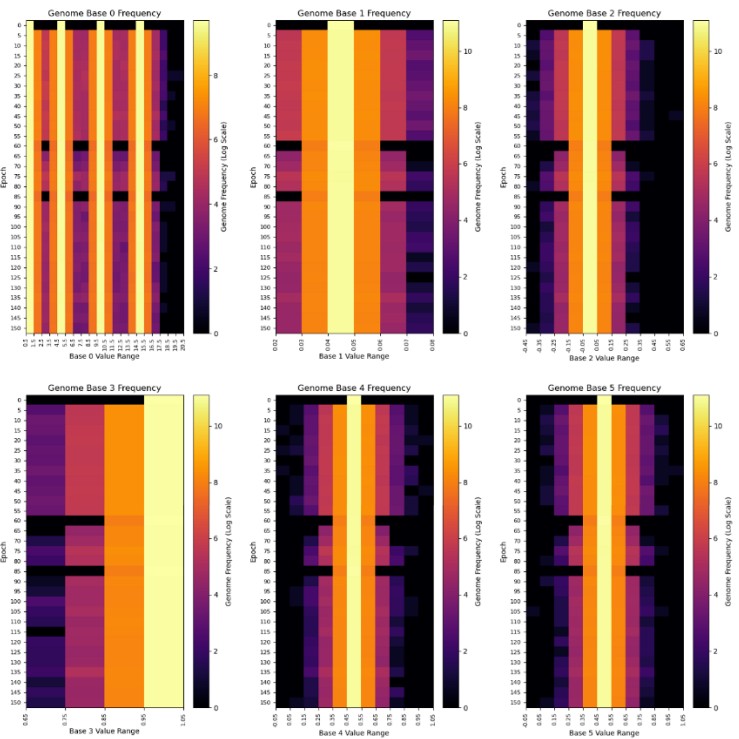

Figure 6: We plot the evolution of genome base distributions over the course of training for a single GCL run. The diversity of each genome base decreases over training.

# F Discussion

## F.1 Limitations

We acknowledge that there are realistic distributions out there that GCL still fails to consistently generate and train on, as noted by worst-case gaps that are still significantly above average.

This paper also focused on instances with 50 nodes. With more compute, GCL could be tested at scale to see if trends hold with larger instance sizes.

Additionally, while we conducted basic hyperparameter searches for curriculum-based parameters, by no means is our search exhaustive due to the large parameter space. Thus, we believe that our performance could be optimized upon further tuning.

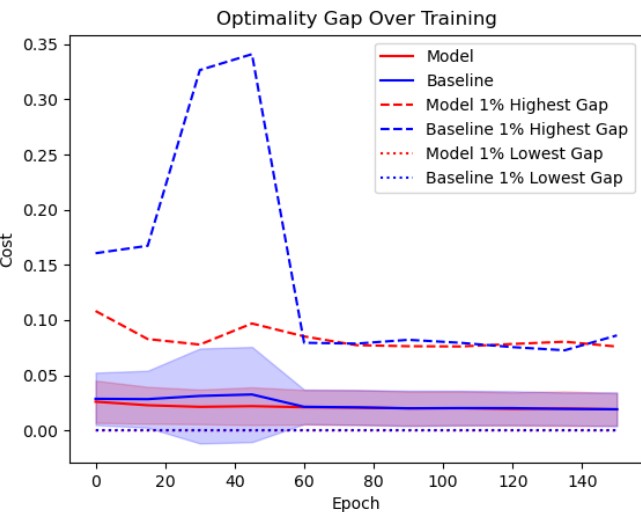

Figure 7: We plot baseline model and current model optimality gap on training data over the course of training for a single GCL run. The baseline gap starts considerably higher than the current model gap, but decreases significantly at around epoch 60.

