# OpenReview forum: "Genetic Curriculum Learning for Distribution Generalization on the Travelling Salesman Problem"
_NeurIPS.cc/2024/Workshop/MATH-AI — MATH-AI 24_

### Official Review · Reviewer_nS2u · 2024-10-07
**Genetic Curriculum Learning for TSP: Promising but Marginal Gains**

**Rating:** 6
**Confidence:** 3

**Review:**

This paper addresses the challenge of distribution generalization in the traveling salesman problem (TSP). The authors highlight the limitations of the baseline method, HAC, which struggles to handle instances significantly deviating from a uniform distribution. To overcome this, they propose a novel approach based on genetic curriculum learning that incrementally trains on harder distributions. Experimental results demonstrate that the proposed method outperforms HAC in these more complex scenarios.

Pros:

The application of genetic curriculum learning to improve distribution generalization is a well-motivated and intuitive extension. It effectively addresses the shortcomings of the baseline method in a structured manner.

Cons:

While the proposed method shows improvement, the gains over HAC appear modest, raising questions about the practical significance of the enhancement in certain scenarios.

---

### Official Review · Reviewer_Qmqg · 2024-10-07
**The GCL method is interesting, but its practicality is in question.**

**Rating:** 6
**Confidence:** 3

**Review:**

The paper uses LLM to solve the TSP, which is well-known to be NP-hard. Previously, LLM and heuristic solvers could do well for small problem sizes, but struggle in larger problems, and real-world data could be out of the distribution of the training dataset. This work proposes a genetic curriculum learning approach to address distributional robustness and a new dataset for more realistic data distribution.

Strengths:
- The proposed approach to address distributional robustness is interesting and could be applied in other settings as well.
- The work has a substantial amount of contributions and results.

Weaknesses:
- Aside from training LLM, the significance of this work is a bit questionable. LLM has a slow inference time and is inconsistent. Moreover, RL-based methods are usually better for this task, so from a solver's point of view this is not very efficient. However, I do appreciate the attempt to improve LLM's capability to solve this task.

- I'm a bit skeptical of the amount of computational resources dedicated to training for this task. The family of genetic algorithms is usually very inefficient, so I doubt that this method would be practical.

---

### Official Review · Reviewer_m57c · 2024-10-07
**A good TSP solver for real-world applications**

**Rating:** 6
**Confidence:** 3

**Review:**

**Summary**

The authors propose a method Genetic Curriculum Learning (GCL) to solve the traveling salesman problem (TSP). They present a neural combinatorial optimization (NCO) approach to solve the TSP problem that improves the generalization and robustness of the model on real-world practical TSP distributions. The authors propose an automatic curriculum in which the environments and tasks are adaptively evolved to be more challenging by mutating training distributions that have high-improvement-potential.
The paper also introduces a new dataset TSPLib50 consisting of 10k TSP instances sampled from realistic distributions.

**Strengths**

The work is shown to improve the performance of TSP solvers on hard and practical distributions.
The authors mentioned statistical significant values for all the experiments they conducted to show improvements of GCL on “worst-case” optimality gaps.
The authors also compared GCL with LLMs (with chain-of-thought reasoning) to solve the TSP problem.
The robustness and generalizability of the approach on real-world practical scenarios facilitates significant real-world deployment and applications.


**Weaknesses**
The authors could have talked about any real-life practical problems/examples to demonstrate the effectiveness of their approach.
It would be nice to compare the results with ChatGPT o1 as well (as it is claimed to have better reasoning capabilities).

---

### Decision · Program_Chairs · 2024-10-09

**Decision:**

Accept

**Comment:**

This paper presents a new algorithm for the classical TSP problem that will be of interest to the MATH-AI community. The authors are encouraged to include more analysis of their algorithm and include a more detailed discussion/comparison of more recent related work such as Sym-NCO.